# *"I Had No Hope, I Had No Help at All"*: Insights from a First Study of Fathers and Recurrent Care Proceedings

**Georgia Philip** [1,*], **Lindsay Youansamouth** [2], **Stuart Bedston** [2], **Karen Broadhurst** [2], **Yang Hu** [2], **John Clifton** [1] and **Marian Brandon** [1]

[1]  Centre for Research on Children and Families, School of Social Work, University of East Anglia, Norwich NR97TJ, UK; j.clifton@uea.ac.uk (J.C.); m.brandon@uea.ac.uk (M.B.)
[2]  Centre for Child and Family Justice Research, Department of Sociology, Lancaster University, Lancaster LA1 4YW, UK; l.youansamouth@lancaster.ac.uk (L.Y.); s.bedston@lancaster.ac.uk (S.B.); k.broadhurst@lancaster.ac.uk (K.B.); yang.hu@lancaster.ac.uk (Y.H.)
*  Correspondence: g.philip@uea.ac.uk

**Abstract:** This article presents data from the first large-scale study of fathers involved in repeat (or recurrent) care proceedings in England. The project complements important research on mothers and recurrence. It consisted of three elements: an analysis of population-level administrative data from the Child and Family Court Advisory and Support Service (CAFCASS), a survey of fathers in pre-proceedings and care proceedings, and a qualitative longitudinal (QL) study of recurrent fathers. Here we report findings from the survey and the QL study, offering an expanded definition and description of fathers and recurrence. Elsewhere, we reported that a significant number of fathers appear in recurrent care proceedings and that the majority return with the same partner. Alongside this, there is also a notable pattern of "missing" fathers demonstrated by the proportion of lone mothers reappearing before the court. Our survey indicates a certain profile of recurrent fathers, but also that recurrent fathers are not straightforwardly a homogenous group. We report on the significance of recurrent fathers' early lives, on the phenomenon of enduring couple relationships and on the prevalence of issues affecting parenting, such as poor mental health, substance use and domestic abuse. Insights from the QL study in particular reveal legacies of harm, loss, and a lack of emotional and relational resources in childhood, which have debilitating and far-reaching consequences. We argue the importance of understanding the vulnerabilities of recurrent fathers and of challenging certain assumptions in child welfare and family justice practices. There is much to be learnt from existing services for recurrent mothers, but also a need for bespoke or adapted services that may be more responsive to particular circumstances of recurrent fathers and couples.

**Keywords:** recurrence; fathers; care proceedings; family justice

## 1. Introduction and Background

Over recent years, in England, the number of local authority legal applications for care orders have reached record levels, although the increase has not been the same throughout the country [1,2]. In England, a local authority may apply to the family court for a care order to consider placing a child in public care, where there is serious harm or serious risk of harm to that child. Care orders, or "public law" applications arise in conjunction with, or following, other local authority interventions such as child protection planning where there are serious child welfare concerns. This increase in care order applications has been occurring alongside a heightened demand for services—for example, increases in the numbers of child protection referrals and child protection plans [3–5]. In 2016, the pattern of increased demand alongside successive funding cuts to child and family services in the UK was

declared by a senior High Court Judge to be reaching critical proportions [6]. The "Care Crisis Review" established as a direct response to these challenges, noted widespread unease about how a lack of public service resources, coupled with poverty and deprivation, were making it harder for both the system and families to cope [2].

The growth in care proceedings in the UK has also raised important questions about public law applications involving parents who had already experienced the removal of children from their care, or who were at high risk of having any subsequent child removed. Such applications have been termed "recurrent care proceedings" [7,8] and evidence suggests that these involved at least 43,500 mothers but also 30,000 fathers between 2007 and 2014 [9]. The work of Broadhurst and colleagues has also provided important evidence to support the growth of grass roots initiatives, such as those discussed elsewhere in this special issue, with the aim of preventing recurrent care proceedings. Such studies have generated a growing body of evidence on a vulnerable and marginalised population of women. Multiple local authorities and third sector organisations in the UK now offer services to address recurrence, with notable examples being the "Pause" programme in England and "Reflect" in Wales, both receiving government funding. However, in contrast to this, very little is known about fathers and the extent, circumstances, and experiences of repeat care proceedings and subsequent loss of their children. This relative "invisibility" arguably sits counter to an increased interest in father engagement [10,11] and broader social expectations for father involvement [12,13]. If there are wider social policy commitments regarding the importance of fathers to children and to the sharing of care, then understanding the life circumstances of highly marginalised and/or disadvantaged fathers must surely be pertinent.

*The Developing Policy and Practice Landscape for Reducing Recurrent Care Proceedings*

The vulnerabilities of recurrent mothers and the multiple or "collateral consequences" of child removal have been key findings from Broadhurst and colleagues [7,14]. Most recently, Broadhurst and Mason [15] have drawn attention to women's enduring histories of disadvantage, as well as cumulative social and economic disadvantages in their adult lives. Whilst there is increasing recognition of histories of disadvantage and unresolved or disenfranchised loss [16], another focal point of policy debate has been how to respond to parents' problems as part of protecting children [17]. Research on mothers' experiences of recurrent care proceedings has strengthened the economic and moral arguments for working with parents' post-proceedings, and for acknowledging the enduring and indeed recurrent nature of problems such as poor mental health, substance misuse and abusive relationships [15,17].

There are also important debates and dilemmas over how to balance the development needs and timescales of children with the timescales for parental change, stability or "recovery". Service developments such as the Family Drug and Alcohol Courts (FDACs) [18] and programmes such as "Pause" have contributed important knowledge and insights here, and also collaborations between partner agencies and parents with lived experience. Key messages have been the need to recognise the enormity of rehabilitation challenges faced by recurrent parents, that services built on trust-based helping relationships cannot be short-term, and that alternative conceptual and treatment models that see recovery as "non-linear" may have much to offer [15,19,20].

In the UK to date, most interventions aiming to reduce recurrence follow a mother-centred approach, involving a holistic and/or long-term service delivered via a trusted key-worker. In addition, and also prevalent in services for birth parents of adopted children, there is specific interest in "trauma informed" work, which aims to develop more directly therapeutic and co-constructed ways of supporting birth parents [19,21]. Whilst such work is much needed, hard won and hugely valuable, there remains a lack of couple-focused, father-inclusive or father-specific services to address the challenges posed and faced by recurrent fathers. The literature indicates that in comparison with mothers, fathers' lives and experiences of care proceedings share important differences as well as similarities. For example, fathers' lives may be characterised by greater transience than mothers', or by particular experiences of precarious housing and work [22]. In addition, fathers may be more able than

mothers to avoid scrutiny or identification by public agencies, making it potentially more difficult to fully and fairly assess fathers where there are serious child welfare concerns.

A key aim of our research is to address significant questions about fathers and their appearance(s) in local authority care proceedings. Reported elsewhere, our population-level analysis showed that whilst there may be a group of "missing" fathers linked to lone mothers in recurrent care proceedings, there is also a significant group of recurrent fathers who are visible to the court. We found that the majority of recurrent fathers reappear before the court with the same partner (79%) and in addition, if the younger recurrent fathers are at the first set of care proceedings, the more likely they are to return. Further to this, age and relationships, both horizontal (partner relationships) and intergenerational (parent–child relationships), are relevant to understanding recurrence [23]. Efforts to address recurrence involve engaging with longstanding and ongoing debates about father engagement, the gendered organisation of care and parenting, and stereotypical thinking about fathers and mothers in public services [24].

## 2. Methods: The "Up against It" Study

Our mixed methods project was carried out between 2017 and 2020 and involved 20 local authorities across England, over a period of 18 months. It consisted of three strands: an analysis of population-level administrative data from the Child and Family Court Advice and Support Service (CAFCASS), a survey of fathers in pre-proceedings and care proceedings and a qualitative longitudinal (QL) study following a group of recurrent fathers over time. There was a geographic spread of participating English local authorities, including the South East, South West, Central, North East and North West regions. There was also a mix of large and small authorities, unitary authorities, London Boroughs and County Councils. In addition, we also partnered with eight voluntary organisations (a mix of national and local) who helped with recruitment for the survey and qualitative elements of the research. The project received ethical approval from both University Research Ethics Committees, the Children and Family Court Advisory and Support Service, and Her Majesty's Courts and Tribunals Service (HMCTS). We also obtained approval for the survey in participating local authorities from the Association of Directors of Children's Services (ADCS) research group, and via local governance processes in some cases. Careful preparatory and ongoing work was undertaken in each local authority to negotiate the fine details of the approach needed locally to administer the survey and recruit recurrent fathers for the QL study.

The overall objectives of the project were to provide the first picture of the scale and pattern of fathers in first and subsequent care proceedings in England, to build a picture of the life circumstances of such fathers, generate insights into the lives and coping strategies of recurrent fathers, and identify opportunities for policy and practice responses and development. The first objective was met through the analysis of population-level administrative data held by the CAFCASS. Analysis of the data provided, for the first time, population profiles of fathers' recurrent appearances before the family court vis-à-vis existing evidence on mothers' recurrence, and we have reported on this elsewhere [23]. The second and third objectives were met through our survey and the qualitative longitudinal (QL) study.

The survey consisted of an anonymous, multiple choice (and one open-ended question) paper-based survey, comprising two parts: one completed by the father (with support where required), and the other completed by a practitioner involved in the child's case, with the father's agreement. The design of the survey drew on previous work by Broadhurst et al. [14] and Brandon et al. [22], the Adverse Childhood Experiences (ACEs) survey [25] and Understanding Society, the largest nationally representative social survey in the UK. Each "pair" of surveys shared an identification code, for matched analysis. Surveys were returned via prepaid envelopes and fathers received a 10 British Pound Sterling gift voucher in recompense for their time. In total, we received 127 surveys completed by fathers and 106 where the practitioner had completed their paired component.

The aim of the qualitative longitudinal (QL) study was to focus specifically on recurrent fathers and gain micro-level insights into their experiences of repeat involvement of children's social care and local authority care proceedings. A QL methodology [26,27] was used in order to develop a prospective study and facilitate insights into trajectories, transitions and turning points, as fathers experienced and attempted to cope with the loss of children and the consequences of this on their lives as men. There is existing scholarship on fatherhood, such as work by Miller [28], Davis and Neale [29] and Tarrant [30], that adopts a longitudinal approach, and which helps to contextualise the different experiences of men's lives as fathers. The QL method also involves thinking about time, in a theoretical sense, in terms of how lives are narrated, remembered, and imagined [31]. The QL methodology therefore provides a device to examine the temporal aspects of the process of living through and with recurrence. The study involved in-depth interviews with fathers (and in some cases couples) and regular monthly phone contacts over a period of between 6 and 12 months. The interviews, at the start and end of the study period, covered all aspects of fathers' histories, lives, relationships and experiences with professionals, and the monthly contacts involved catching up and reflecting on any changes over time. Twenty-six fathers from across England took part in the QL study, with a high retention rate—we have complete or almost complete data for 23 recurrent fathers.

## 3. The Study Sample(s)

The study was designed to investigate fathers and recurrence at a macro, meso and micro level, with each element addressing corresponding but related research questions. The aim was to capture a population-level picture of the scale and pattern of fathers and recurrence, a descriptive picture of recurrent fathers as a group, and a deeper understanding of recurrent fathers' experiences. However, it is important to note that our samples are substantively rather than methodologically nested. Specifically, whilst the sample of recurrent fathers in the QL study are not a subsample of the surveyed fathers, the survey nonetheless provides a wider context in which the QL recurrent fathers can be located. In turn, whilst the survey provides broader significant characteristics of recurrent fathers, the smaller QL sample enables more nuanced information and insights on their life trajectories and lived experiences over time. This substantive and iterative approach that we used to build our survey and QL samples was largely due to the recruitment challenges we encountered and the need to be responsive, flexible and sensitive to the circumstances of both the local authorities and the fathers involved.

## 4. Defining Fathers and Recurrence

Determining men's parenting status is complex, involving biological, social and legal factors. We took an inclusive approach to defining fatherhood to acknowledge the range of fathering roles and relationships men may experience over time and the ways in which these mediate their involvement with children's social care and the family court. Our main inclusion criterion was being a biological father to at least one child subject to local authority care proceedings but the majority of men in the QL study were also, or had been, stepfathers and/or mothers' partners involved in the first or repeat care proceedings.

Defining recurrence is also more complex than it first appears. In our study we built on the original definition of recurrence coined by Broadhurst et al. [14], which distinguished between recurrence with at least one new child, or recurrence with a previous child or children only. For our work, we also added that a parent may return with the same partner, a new partner, or with no identified partner [23]. For the survey there were questions to determine previous involvement in care proceedings and whether the participant had children living in out-of-home care, plus the possibility of matching responses between fathers and practitioners. For the QL study, where we specifically sought to recruit recurrent fathers, we defined recurrence as having had two or more experiences of any combination of pre-proceedings, care proceedings, or voluntary accommodation (S20) of children, rather than simply two or more instances of child removal through Section 31 care proceedings. This iterative

decision was taken in order to accommodate the range of contexts in which fathers experienced the loss of children and the range of outcomes of care proceedings including Special Guardianship Order, Supervision Order, or being placed in the father's care [32]. The wider definition was also employed to overcome the difficulties we might otherwise have encountered in recruiting sufficient participants in the time available.

## 5. Key Findings from the Survey on the Life Circumstances of Recurrent Fathers

Overall, our findings suggest caution about perceiving recurrent fathers as a homogenous group and indicated more similarities than differences between recurrent and non-recurrent fathers. There was a level of homogeneity across all fathers involved in care proceedings, in terms of being predominantly White British, English speaking, non-religious, never-married, becoming a father at a younger age, and living in economic hardship. The factors that pertained significantly for recurrent fathers were being looked after as a child, experience of multiple childhood adversities, being unemployed, and not living with their youngest child.

Of the 127 father surveys, 70 (61%) were identified as recurrent by having previously experienced care proceedings and having at least one child living in out-of-home care as a result. In relation to their own experience of out-of-home care, just under a quarter of recurrent fathers from the survey (22%) had been looked after as a child. In relation to childhood adversity, around half had experienced abuse and/or neglect in childhood, along with parental separation (parental separation was prominent among non-recurrent fathers also). The reported experiences of recurrent fathers highlighted the impact of multiple rather than single adverse experiences, with family- or household-related factors including family conflict and violence as well as family drug misuse and poor mental health. Our analysis of the ten ACE questions indicated that the majority of fathers surveyed (63%) had experienced "single issue" adversity, primarily parental separation. The remainder (37%) had one of two "multiple issue" profiles, one characterised by direct physical abuse and/or abuse between parents and the other characterised by parental/family mental health issues and/or parental or family substance misuse. We found that recurrent fathers were significantly more likely to fit one of the multiple issue profiles (51%). However, it is important to note that just under half the recurrent fathers did not experience such multiple and extensive adversities.

Overall, findings from the survey demonstrated that recurrent and non-recurrent fathers substantially overlap in their economic disadvantage. Over half the fathers surveyed were either unemployed (43%) or were economically inactive due to long-term sickness or disability (13%). In line with this, we found high levels of welfare benefits (e.g., Universal Credit, Income Support, Tax Credit, Housing Benefit, and Job Seeker's Allowance) being received by either the father or someone else living with them. The only significant distinguishing factor for recurrent fathers was a higher level of unemployment (69%).

Although recurrent fathers were less likely to be living with their youngest child, the survey showed no significant difference between recurrent and non-recurrent fathers in terms of contact with children they did not live with. The majority of all fathers surveyed appeared to have some level of contact with at least one/some of their children. A minority said they had no contact with their youngest child (12%) or contact a few times a year (7%). These findings prompt certain questions about father "absence" and indicate the relevance of understanding father involvement across households and over time.

In addition to the characteristics that did significantly pertain to recurrent fathers, it is relevant to note other factors relating to recurrent and non-recurrent fathers alike, in part because they raise equally interesting questions for practice. For instance, a large family size was not a significant characteristic, and recurrent fathers were no more likely to have large families than non-recurrent fathers. The survey showed that the majority of all surveyed fathers had one or two children, and a minority had three or more.

In terms of fathers' health and wellbeing, the survey indicated that physical, mental and emotional health issues are relevant for both recurrent and non-recurrent fathers. When asked about longstanding physical and mental health issues over the previous 12 months, 14% of all surveyed fathers stated they had physical health issues, 20% had mental health issues, and 16% reported having both mental and physical health issues. Fathers were also asked about emotional stress and here reporting was higher, with just over half (53%) stating that their daily lives were affected by emotional stress some or most of the time.

In terms of the reported child welfare concerns, the 106 matched and completed practitioner surveys allowed some measure of what the primary concerns were and which (if any) related directly to the father. Findings showed that neglect (71%), emotional abuse (48%), and physical abuse (35%) were more prevalent and sexual abuse was identified in a minority of cases (10%). In terms of child welfare concerns related directly to fathers, domestic violence was the most common (identified in 52% of cases), followed by drug misuse (37%), separation/relationship breakdown (27%), and poor mental health (26%). Our analysis of the survey data also produced an exploratory typology of fathers and child welfare concerns, indicating the increased impact of multiple or cumulative problems affecting men's parenting.

Overall, findings from the survey suggest that recurrent fathers, whilst sharing certain characteristics with fathers appearing in care proceedings for the first time, are more likely to have experienced cumulative adversities in their early lives, to have been looked after as a child, to be economically disadvantaged, and to have never lived with their youngest child. Recurrent fathers also appear likely to have, often multiple, health vulnerabilities, including, or related to problems with substance misuse and domestic abuse.

## 6. The Characteristics of the Recurrent Fathers in QL Study

Whilst the purposive sample of 26 recurrent fathers who participated in the QL study was not directly linked to the survey, it did reflect the significant characteristics of recurrence outlined above. The QL study generated rich information and insights about men's lives and relationships with their own parents, families, partners and children. It also builds a dynamic picture of the fathers and the processes of recurrence.

The 26 recurrent fathers ranged in age from 23 to 51 (average age 34.5), were predominately of White British ethnicity, had entered fatherhood relatively young (five had become fathers under the age of 20), and were living economically and socially precarious lives. All had experienced multiple local authority interventions including pre-proceedings and care proceedings. In addition, most had also lost children from their care due to private law proceedings, separation, divorce and family estrangement.

The majority of the fathers described pronounced or extensive childhood adversity, having experienced maltreatment in relation to physical, sexual and emotional abuse arising from their immediate or wider familial network. In terms of physical, mental and emotional health across their life course, there was also evidence of cumulative and/or pronounced levels of problems experienced. Issues included depression, anxiety, insomnia, chronic health conditions (such as back problems, asthma or epilepsy), and ongoing substance misuse problems with cannabis and alcohol being most common. In addition, just under half had some sort of diagnosis of learning disability, cognitive or behavioural condition, including ADHD, Autism or Asperger Syndrome. Fathers' and/or couples' learning disabilities were also notable as a significant factor directly related to the removal of children into local authority care (there were five fathers for whom this was the case).

In terms of child welfare concerns associated with these 26 recurrent fathers, the picture was again complex and dynamic. This complexity arises through histories of past and recent experiences of local authority involvement, because crises occurred, or reoccurred, with different partners or households, and because the focus of concern was not always, or exclusively, on the father. Neglect, often in connection with emotional abuse, were factors relating to just under a third of the sample. Two fathers had been found to have physically harmed or posed a risk of physical harm to a child in their care.

Three further fathers were deemed to pose a potential sexual risk to a child, arising from having been charged for a sexual offence and/or having experienced sexual abuse themselves in childhood. Substance misuse, domestic violence concerns and poor mental health were also prominent, with half of the sample experiencing child protection and care proceedings due to some combination of these factors. Poor mental health and substance misuse appeared to have been longstanding or recurring problems for both fathers and mothers. Domestic violence concerns had been raised about fourteen fathers, and these included the perceived risk of domestic abuse due to past history, allegations and counter-allegations, and in two instances recognition of couple violence. Just over half the recurrent fathers (16) had some sort of offending history, with six having been in prison at some point in their lives. Seven fathers had been either cautioned or charged with a domestic violence offence, but none had been imprisoned for violence against their partners.

## 7. Understanding the Lives of Recurrent Fathers; Two Themes from the QL Study

In the following section, we present two prominent themes from our analysis of the QL data in order to offer illustrative evidence of the complexity of recurrent fathers' lives and the relationship and parenting problems they pose and face. Our QL data were analysed using the "Frameworks" approach [33] involving indexing, sorting and summarising in two key ways—case and wave. This involved analysing and comparing each recurrent father's individual "case" and also chronologically comparing across the whole sample by time wave. This approach supported the longitudinal focus on change over time and, specifically, transitions, turning points and trajectories [26].

## 8. Theme 1: The Early Lives of Recurrent Fathers

Our longitudinal approach involved focusing on the life histories of recurrent fathers, in keeping with existing research on mothers' experiences of recurrent care [15]. A powerful overall theme from the QL study, was of fathers with unresolved childhood trauma that appeared to blight their capacity for emotional regulation, nurturing relationships and family functioning.

Childhood neglect, abuse, parental separation, estrangement from caregivers, parental mental ill health, domestic abuse or substance misuse were prominent in these recurrent fathers. Their early lives were also marked by further forms of instability including frequent house moves, bullying at school and disrupted education. A lack of felt physical and emotional security was particularly evident among the men who had experienced abuse. For example, Keith's biological father was extremely violent towards him from a very young age: "he put me in hospital about five times". When Keith was two years old, he was taken into care. At the time we first interviewed him, Keith had recently found out that he lived with five different foster families before being adopted at the age of seven. Keith never felt that he fitted in with his adoptive family. He continued to experience physical chastisement at the hands of his adoptive father who "smacked [him] on the back of [his] legs". At the same time, Keith experienced a lack of emotional connection with his adoptive mother, which continues to impact on him as an adult: "there was no love. What you would want from your adoptive mother is love, show love, I am missing that now".

A second difficult example comes from Matthew's story. Matthew's parents separated when his biological father was sentenced for the sexual abuse perpetrated against him and other children in the family network. However, his parents' separation and father's imprisonment was not the end of the collateral consequences, which continue to haunt Matthew's family (to the present day). Matthew described how his mother felt obliged to move when people found out that his father was a paedophile, and the family could not settle due to the public shame of his father's offence. Tew [34] discusses how the incidence of trauma may incite public and private shame, as the family identity is spoiled, which in turn can result in wider social disconnections, even when a perpetrator is excluded from the family. Matthew's story illustrates the reverberating impact of traumatic events on the lives and functioning of him and his whole family and over many years.

Fathers who felt abandoned or rejected, by one or both their parents, were often left feeling enraged, hurt, humiliated and/or scared, emotions which they said often led to further harmful behaviours. Stories of adolescence characterised by "falling in with the wrong crowd", developing drug or alcohol dependence, and episodes of poor mental health were also notable. Travis' parents separated when he was six, following which he continued to live with his mother until her death when he was 13 years old. After the profound loss of his primary caregiver, Travis found himself bounced between his older brother, his cousin and his father's care, until his father asked him to move out following arguments with his stepmother. At the age of 15, Travis found himself isolated from his family and living on his own. The experience of having to either leave home at a young age without support or having insecure or unsafe accommodation was noticeable, as was the pattern in their adult lives of housing being directly linked to the making and ending of couple relationships.

Whilst there was powerful evidence of insecure or destructive early life relationships for recurrent fathers in our study, it is also important to identify the presence of reliable or protective figures in some men's lives. Of note was the presence of grandparents who took on caring roles and continued to "be there" for men as they faced ongoing or escalating problems. However, when their grandparents die, the loss to men in our study could be profound. Examples of men who were distraught by the deaths of a grandparent include Michael, Jonathan, Jack, Danny, Joe, Martin and Brian. For example, Joe recollected the intense impact his grandad's death had on him personally, as well as on the wider family dynamics.

> When my grandad died I felt a bit suicidal … He were the structural foundation of our family, as soon as he died our family has fallen to shit, pardon my French but it did, we have all grown apart, we all fall out … He was the glue that binded the group together. (Joe)

The impact of adverse early childhood relationships, particularly those relationships characterised by maltreatment, reverberated throughout childhood, into adolescence and adulthood, as these fathers reflected back over their lives. Turbulent and abusive early childhood experiences left men feeling confused, let down and impacted on their sense of confidence and self-esteem. From a young age, many of the men had learnt that they could not trust others to meet their needs or keep them safe. In turn, adverse relationships in childhood, between parents or across the extended family, appeared to predispose men to further adversities, including unstable housing, educational difficulties, mental distress and substance use. Bereavement and/or abandonment by parents often resulted in some men finding themselves out in the world on their own whilst still children. Substance use beginning in childhood presented a strategy to lessen the emotional pain of feeling abused, abandoned or rejected. In addition, using substances provided a pathway into being "accepted" by peers, where some (young) men found a sense of belonging. A lack of stability or parental care, in combination with feelings of isolation and rejection, appeared to have contributed to some men falling into a chaotic lifestyle of drugs, drink and sex.

There were a number of situations or points in their younger lives where these men felt that they needed and lacked both practical and emotional support. A lack of support at the time failed to mitigate harm, and some fathers found themselves caught in a "vortex" of negative or destabilising change in their lives that they could do little to control. Some then described feeling increasingly judged by their pasts and faced what they saw as repeated or inevitable punishment for their actions. In particular, and by their own accounts, many recurrent fathers felt that as they entered couple relationships and became fathers they did so with few emotional resources, or with emotional coping strategies that ill prepared them for intimacy and parenthood.

> From other people's perspective, they just see me go from one relationship to another to another to another, to be fair I get that is how it looks on the outside … Do you know what, it is a fear to be on my own. It is fear, I am not going to lie. (Mark)

## 9. Theme 2: The Significance of Couple Relationships

Relationships with partners were of great emotional, psychological and practical significance to the men in our study, as individuals but even more so as fathers. The role of mothers in mediating father-involvement, and the significance of the co-parenting relationship (as distinct from the couple relationship) is well-recognised in the wider context of family life [35,36]. There are also longstanding and ongoing anxieties over father absence [37]. Stereotypical descriptions of "feckless fathers" (in tandem with negative stereotypes of lone mothers) have depicted some men as having numerous children to multiple women, avoiding financial and moral responsibility. This rhetoric is not specific to the UK. In the U.S. those fathers who do not fulfil their parental responsibilities have been pejoratively termed "deadbeat dads" [38].

The stories of recurrent fathers from our QL study revealed men's relationship histories and dynamics, demonstrating ways in which family ties and affective bonds between adults and with children operated across households, geographic distance and time. Divorce, separation, non-resident parenthood and "second-family-hood" were features of recurrent fathers' lives in addition to the material, personal and relationship challenges which brought them into child protection services and the family court. The study contained many stories of relationships with ex-partners and ongoing parenting or co-parenting relationships. Whilst couple conflict and conflict between ex-partners was prevalent, it is important to also note continuity or at least the dynamics of fathers' involvement with birth children and stepchildren not in out-of-home care over time. Our findings highlight the importance of challenging assumptions about recurrent fathers' non-involvement or "absence". They also demonstrate the need to be curious about fathers' lives and relationships beyond the immediate circumstances and time frame of any given case.

Over a third of recurrent fathers in our QL study were in longstanding and what we termed enduring relationships. The remaining fathers were separated and recently repartnered or were single. The length of such enduring relationships ranged from 4 to 25 years, in part dependent on the age of the couple at the time of our study (younger enduring couples tended to have been together for fewer years). The histories and circumstances of enduring relationships varied but the key element was the pattern of couples experiencing long periods of children's social care involvement and public law proceedings together. Characteristics of enduring relationships included either the man or the woman having had previous relationships (and children), and one or both having children in out-of-home care before they formed a relationship. We also use the term enduring relationships to describe those fathers who separated and reunited with the same partner over a number of years. For these men, despite having had separations and/or short-term relationships with other women, their partner status remained primarily with the same woman who was the mother to at least one shared biological child, over a number of years. Examples of enduring couples include Jonathan and Megan, Keith and Emma, Michael and Kath, and Tony and Dawn, who are all now in their thirties and forties.

Me and Dawn are There for Each Other Really, if She's Down I Lift Her up, and She Does that for Me. (Tony)

Jonathan and Megan's relationship offers a particularly poignant example of multiple and interconnected factors leading to their repeat experience of child protection processes and care proceedings over a period of fifteen years. Jonathan is estranged from his family, has chronic health issues, and struggles with his mental health. Megan suffers from depression and anxiety and experienced post-natal depression after the birth of their first child. They both have a long-term cannabis addiction. Megan's post-natal depression led to the initial involvement of children's social care, and after the birth of their second daughter, both young children spent time in foster care. Concerns were expressed over the couple's mental health, their cannabis use and Jonathan was seen as controlling and emotionally abusive. The family were reunited after six months and Jonathan and Megan had two further children, with fluctuating interventions from the local authority. The couple's cannabis use, Jonathan's anger and Megan's mental health remained concerns. Then, following a family crisis when their eldest child

Mia's behaviour became increasingly challenging, care proceedings were initiated and resulted in a long-term foster placement for Mia. The remaining children continued to have a child protection plan and as part of this the couple were advised to separate and to work on their own issues. They did separate but got back together without the local authority's knowledge. When this came to light, second proceedings were started for the three younger children, and they were all placed in foster care. The outcome was long-term foster care for all three, involving separate placements. Jonathan and Megan have continued to separate and reunite, though have not lived together again. Jonathan has actively sought help for his mental health and cannabis addiction from voluntary organisations; he continues to progress and relapse. Megan appears to have become more isolated and struggles with her mental health and cannabis use. Both parents have some supervised contact with their younger children.

Jonathan described the couple's conflicted emotions, in relation to the social worker and to each other, during the second episode of pre-proceedings and then care proceedings:

> She [the social worker] told Megan 'well if you get rid of him you will get your kids back faster' and at the time I went to stay with a friend for a few weeks, but in the end Megan went 'you are their dad at the end of the day you need to be here so fuck what the Social Worker is saying if it takes a bit longer it takes a bit longer but I can't do this on my own.

Jonathan and Megan's experience illustrates a feature of enduring couple relationships that we described as couple "jeopardy". This involved situations where a couple was faced with the decision to separate (or not) as part of the case management approach taken by the local authority. Some couples attempted to strategise, hoping that mothers were more likely to keep children in their care if they separated, but others found this prospect too overwhelming in terms of parenting challenges, or too painful. As Jonathan and Megan and also Michael and Kath experienced, reuniting as a couple is a high-stakes decision and the outcome is often devastating:

> Because me and Kath wanted to get back together again after we had split up a bit, you know, we decided we wanted to try again, but they decided if you are going to try again you are going to do it without the kids and that was pretty much it. (Michael)

This theme sheds light on fathers' perspectives on child protection and care proceedings, where the primary focus is children's welfare. Whilst the prioritising of children's timescales and needs may be seen as non-negotiable, it is nonetheless crucial to understand the experiences of fathers as partners and as parents. In the context of recurrence, it is also relevant to consider how local authority approaches that involve the separation of couples may impact on fathers but also on mothers and on children in different and particular ways. The findings from our study suggest such approaches could have the effect of placing unrealistic expectations on couples, overburdening mothers, and overlooking or excluding fathers from care planning processes. More widely, it is recognised that focusing on mothers also has implications for the paternal family network and the potential resources and relationships this may provide for children. The impact on recurrent fathers of the passing of time and the legacies of intimate relationships and relationships with professionals is also notable and relevant. As part of his final interview, Jonathan reflected on his enduring but painful relationship with Megan. His comments here illustrate the dilemmas involved in his sense of responsibility as a father, and his intense emotional bond with Megan as a partner.

> Would you still like to get back together?

> Yeah the thought is there but, Jess is sixteen next year, if she were to suddenly turn round and say 'oh mum can I come back to you?' she wouldn't come back knowing that I was still on the scene and I can't let that happen so . . .

And I have told Megan so many times in the past few months I need to move on, and I have tried walking away and it has been me that has been phoning up her. But lately, it has been like a few days and I get a text from her saying 'I need you back' so, it does make the situation a bit harder.

As indicated through the two themes presented, whilst enduring relationships were a feature of recurrence for fathers, these were often fraught with difficulties that exacerbated couple conflict and escalated professional concerns. Despite the longevity of some relationships, and the feeling that problems were "theirs", recurrent fathers and their partners often felt that services did not work with them as a couple. Whilst there is a prevailing interest in whole family approaches and practice initiatives such as Family Group Conferencing, recurrent fathers and their partners (some of whom were recurrent mothers) notably perceived the local authority focus to be on mothers or on them as individuals. Fathers and couples in our study who stayed, or who wanted to stay, together often felt that this was rarely a possible or supported option, or was one which could be suddenly withdrawn with irrevocable consequences. Couplehood could shift from being seen by the local authority as a demonstration of commitment to one of failure to prioritise and be protective of children's needs.

Recurrent fathers' accounts also illustrated complex ways in which the mental health and functioning of couples was interconnected and experienced together rather than being an issue located within one individual. Similarly, couples' histories of using and misusing substances and of couple conflict were often shaped by social (and material) factors and forms of co-dependence, which were not necessarily acknowledged or responded to by local authority processes. All of these issues had a hugely damaging and destabilising effect on parenting. For recurrent fathers, and their partners (some of whom were recurrent mothers), such problems could then be compounded following the conclusion of care proceedings. This could effectively leave their social and emotional resources and capacity to make changes even more depleted: "I had no hope, I had no what do you call it err 'help', I had no help at all" (Keith).

## 10. Discussion: What Are Some of the Practice Implications of Our Findings?

We have reported on the significance of recurrent fathers' early lives, on the phenomenon of enduring couple relationships and on how parenting problems related to poor mental health, substance use and domestic abuse are experienced by recurrent fathers in couples. The findings presented here and in our population-level analysis [23] show that relationships are particularly relevant to understanding fathers' experiences and trajectories into and out of recurrence. If recurrence is a relational problem, then we argue that the response must also engage fully with and attend to relationships, in all their complexity.

Our findings on fathers' early life experiences do suggest an association between childhood adversity and fathers' appearance in repeat care proceedings, but this does not constitute a straightforward causal link. Again, similarly to Broadhurst and colleagues' [14] findings on recurrent mothers, our observation is that adverse relationships in childhood appeared to predispose men to further adversities, not least by depleting their material, social and emotional resources. The concept of cumulative adversity or disadvantage in relation to educational achievement, housing, employment and health and wellbeing is therefore also relevant to understanding fathers' experiences of recurrence.

Our findings also speak to existing research on developmental trauma and on the importance of early caregiving relationships to emotional regulation and resilience. This field of research has been increasingly explored and utilised in relation to developing services for mothers who have children adopted and/or who have experienced recurrent care proceedings [17,19], but again, there is little complementary research on fathers [39]. The rich life histories and accounts from recurrent fathers in our QL study indicated the emotional and relational vulnerabilities of recurrent fathers, the debilitating, often devastating consequences of this for their relationships, and the lack of sources of appropriate support at key points in their lives (including during and after care proceedings). Alongside the link to "trauma-informed" practice that is gaining ground in the context of post-proceedings support, we also

propose the relevance of concepts of emotional scaffolding [40] and mentalisation [41]. We found these ideas to be applicable and insightful for working with recurrent fathers because of their combined emphasis on the foundational, fundamental, but also restorative nature of our relationships with others. Our observation is that without resources and support to manage emotions and relationships differently, couple conflict and its impact on parenting could be a factor in some families that becomes stuck in a cycle of recurrence. An approach which includes support with emotional regulation and capacity to be both self-reflective and responsive to others may be particularly valuable for recurrent fathers. We suggest this may be in relation to working with individual unresolved past pain, for improving couple dynamics, but also in relation to navigating relationships with professionals [42].

We have also presented findings that highlight the enduring or indeed recurring nature of problems affecting parenting and relationship functioning—most commonly poor mental health, substance misuse and domestic violence. In part, this demonstrates the enormity of the recovery challenge for recurrent parents [15,19] and so strengthens the argument for long-term support as an "active ingredient" in services to reduce recurrence. Our study complements that of Broadhurst and colleagues [14,15] in that we highlight the multiple and cumulative adversities faced by recurrent fathers and the long(er)-term support needs that follow. In addition to the clash of children's and parents' timescales that already exists within the context of child protection and care proceedings, where parents are required to address addiction or chronic mental health issues, the time needed to make realistic or sustainable change is difficult to accommodate [43]. The development of the Family Drug and Alcohol Courts (FDACs) and of post-proceedings services for recurrent mothers, are perhaps examples of where innovative and longer-term approaches to recovery can be seen. However, the point remains that the vast majority of post-proceedings support programmes to date simply do not work with couples or with fathers.

Our research also illustrates further limitations of a continued practice focus on mothers. It may be that there is an ongoing tendency for risk aversion in relation to working with fathers generally. This can be heightened in the high-stakes context of recurrence, and even more so in relation to domestic violence and abuse. However, the argument for more nuanced "high support and high challenge" models for working with men and with couples is significant [44–46]. In the policy and practice context there is acknowledgement of the need to respond to this complexity, which in part has meant reflecting critically on the binary model of victim and perpetrator [47]. There is growing appreciation of the limitations of approaches that may over-responsibilise mothers (Featherstone and Peckover, 2007) and fail to hold fathers' accountable [48]. However, the question of whether violent and abusive men can change and whether abusive partners can be safe or good-enough fathers remains highly political, emotive and contested [49,50]. Programmes that involve a generative and/or restorative approach in order to address the challenge of holding men accountable whilst still engaging them "as agents of change" [44] we would argue, are vitally importance here. More established programmes such as Caring Dads, or new ones like "For Baby's Sake" [45] are examples of strengths based (or non-deficit) and couple-inclusive approaches to changing fathers' abusive behaviours, aiming to harness the motivation to change through men's identity as fathers. However, initiatives such as this are still small in scale and geographical reach and so are unlikely to be available for most parents.

Lastly, in terms of implications for enabling recovery and change, the findings from our study powerfully illustrated that recurrent fathers were living with significant limits on what they could achieve as parents and as partners. Recurrent fathers had few and fragile resources with which to rebuild or sustain relationships with children they did not, or could not, live with. Importantly, whilst there was evidence of positive change for some, there was no one temporal pattern that fitted all cases—recovery, change and growth for these recurrent fathers was not a linear process. The insights gained from the QL study also highlighted how untidy the change process can be. We saw apparent false starts, ambivalent or unwilling attempts to change, pauses, interruptions, relapses, overpromising, failures, and trying again. Whereas the damaging impact of "chaotic" life changes for recurrent mothers has become a focus for professional intervention, we suggest that little attention has been given to

fathers caught up in a similar process. Just as there is an emphasis on "slowing down" or taking a "pause" in order to process past pain and imagine a better future, for mothers, our study indicates that a similar ethos may be equally relevant for fathers.

It is also important to note that reclaiming a father identity and even "doing" fathering is not always or only about direct contact. We suggest that for some recurrent fathers, the generative potential of fatherhood and the options for contributing to children's lives could be appropriately and positively explored in terms of meeting children's identity and family history needs, future financial support, providing stories, explanations or apologies as part of a restorative process [51]. One further implication from our research is about recognising the range of ways in which fathers permanently separated from their children might be supported to provide better outcomes for those children.

## 11. Conclusions

Our research has generated, for the first time, specific insights for understanding and responding to fathers involved in repeat care proceedings. We have presented findings from the survey and QL elements of the project and there are limitations to note. The size of our survey sample (127) means we were not able to conduct advanced statistical modelling. To mitigate this potential risk, the analysis on the survey component of the project was accompanied by an appropriate power analysis to ensure that the analysis we could conduct was robust. Secondly, for both the survey and the QL study, the sample may constitute less marginalised fathers. This is because, in order to be recruited, they had to be visible or engaged with some kind of assessment or support service. It remains the case that the voices of the most marginalised recurrent fathers are likely missing. Our QL study, whilst offering rich and powerful stories of recurrent fathers' lives, does prioritise those fathers' perspectives, and cannot establish any settled "truth" about any given case—this was not the aim. That said, exploring and revisiting events with fathers over time, talking to fathers and their partners together, and noting certain factual elements to each case (such as meetings, hearings, placements and so on) added cross-referencing points that contribute to the insights gained. These full and detailed accounts of recurrent fathers' material, relational and emotional lives are, overall, timely and hugely valuable.

We suggest that working with recurrent fathers requires professional curiosity, a holistic approach, and time to understand their relational histories. It also requires a willingness and confidence to hold the combination of risks and resources that most present, and the rehabilitative challenges they are up against. Attention needs to be paid to the potential differences in public and professional empathy towards recurrent fathers, and the corresponding difference in opportunities for accountability and rehabilitation. Whilst we suggest there is much to be learnt from existing services for recurrent mothers, there is also a need to explore bespoke services, or adaptations of programmes that might be more responsive to the particular needs or circumstances of recurrent fathers. Examples could include, developing approaches to work with couples, focusing on emotional regulation and building emotional and relational resources, and exploring the generative potential of fatherhood as a mechanism for change and accountability. We suggest that there can be a more gender-sensitive approach to understanding and responding to recurrence, and indeed to working with fathers more generally. Our position, and central to such an approach, is a commitment to gender equity in relation to parenting roles and responsibilities. Supporting fathers and mothers cannot be seen as a zero-sum game, where service development for one necessarily diminishes or sits in opposition with the other. In relation to recurrence, we are arguing for the development of services that hold men equally accountable for the safe care of children and avoid positioning women as disproportionately responsible for children's welfare. Such services are urgently needed and require sustainable resourcing, not least in terms of time. Our work highlights the many practice dilemmas involved in balancing such accountability with support and validation for recurrent fathers who pose and face enormous challenges as parents and partners. Such dilemmas are arguably being engaged with by practitioners conducting holistic, assertive and empathic work with highly marginalised mothers. In order to respond comprehensively to the recurrence, an equivalent investment is required for fathers. Without this, highly marginalised

recurrent fathers, their families, and the practitioners trying to support them are likely to remain up against it.

**Author Contributions:** All named authors were involved in the funding acquisition, research design, implementation and data analysis from this project. M.B. and K.B. were Primary Investigators for the project and G.P. was the project manager. G.P., L.Y. and J.C. were primary researchers for the QL study. S.B., Y.H. were primary researchers for the population level study and the survey. For this article G.P. produced the original draft and all other authors have contributed to reviewing and editing. All authors have read and agreed to the published version of the manuscript.

**Funding:** This research was funded by the Nuffield Foundation, grant number SPI/43084.

**Acknowledgments:** We are very grateful for the support shown by our advisory board, who shared our enthusiasm for this challenging research. We are grateful to Cafcass (Child and Family Court Advisory and Support Service) for partnering with us and to all the local authorities who took part. Finally, our special thanks are due to the fathers who took part in the qualitative longitudinal study. These men generously allowed us to walk alongside them for many months and shared their experiences of family and parental life, child protection services, care proceedings and child removal.

**Conflicts of Interest:** The authors declare no conflict of interest. The funders had no role in the design of the study; in the collection, analyses, or interpretation of data; in the writing of the manuscript, or in the decision to publish the results.

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
