# Peer review of "“I Had No Hope, I Had No Help at All”: Insights from a First Study of Fathers and Recurrent Care Proceedings"

_societies, doi:10.3390/soc10040089_

Round 1

Reviewer 1 Report

This extremely well-written and structured article presents insights from a novel, large-scale national study of father involvement in repeat or recurrent LA care proceedings in England. As stated, it addresses a key gap in evidence in its focus on fathers and complements wider understanding of mothers’ experiences of such processes. The welfare concerns of fathers, in relation to women are carefully addressed and placed effectively within the developing policy and practice landscapes for reducing recurrent care proceedings. The methods (including analysis) are robust and the findings well described and justified overall. The data presented is also really rich and insightful and demonstrates the complexity of family lives for those who come regularly into contact with the care system. This paper is an important contribution and has potential to inform several interdisciplinary bodies of scholarship.

As a minor point, I think that the section on the policy and practice landscape could reference and link more specifically to literatures that evidence broader policy shifts towards encouraging father involvement and engagement more generally, to position the experiences of marginalisation and disadvantage of these men within a broader context of fathering engagement.   

In the section on QL methodology, the authors might consider citing QL research and methodology that has increasingly been employed in fathering research, moving beyond ‘snapshot’ views to develop intimate moves of the lives of fathers instead. The work of Tina Miller springs to mind here (2010), as well as the Following Young Fathers study directed by Neale et al. (2015) whose findings also illuminate the experiences of a sub-sample of young men who come into contact with care proceedings. This will again link and position the study within a wider set of methodological approaches being developed in fatherhood research.  

I would suggest providing a definition of the care orders described on page 5 for an international for an international audience. A footnote or citation here would suffice.

On page 5, ‘the’ is missing from the sub-heading about the QL study.

On line 264 of that page I think the authors could add that the QL study builds a dynamic picture of the fathers and processes of recurrence. This is in line with the epistemology of QL research and a rationale for exploring recurrence using these methods more generally.

Author Response

We would like to thank Reviewer 1 for their positive and encouraging comments. We have addressed the specific points raised below:

As a minor point, I think that the section on the policy and practice landscape could reference and link more specifically to literatures that evidence broader policy shifts towards encouraging father involvement and engagement more generally, to position the experiences of marginalisation and disadvantage of these men within a broader context of fathering engagement. 

Response: This was a valid and very helpful point. We have revised the ending of the final paragraph of the Introduction/Background, from lines 53-58 to respond to it and add appropriate references. All additional refs have been included in the reference list at the end. 

In the section on QL methodology, the authors might consider citing QL research and methodology that has increasingly been employed in fathering research, moving beyond ‘snapshot’ views to develop intimate moves of the lives of fathers instead.

Response: Again, and important cross reference to make, to locate our work within this broader approach to studying fatherhood. We have added a sentence to the description of the QL study in the methods section - lines 145-147. All additional refs are included in the reference list.

I would suggest providing a definition of the care orders described on page 5 for an international for an international audience. A footnote or citation here would suffice.

Response: Agree, the clarification is important. We have added a ref to guidance produced by the UK Family Rights Group. Line 193.

On page 5, ‘the’ is missing from the sub-heading about the QL study.

Response: The missing word has been added. Line 266.

On that page I think the authors could add that the QL study builds a dynamic picture of the fathers and processes of recurrence. This is in line with the epistemology of QL research and a rationale for exploring recurrence using these methods more generally.

Response: We have added a short sentence to make this point - lines 270-271

Reviewer 2 Report

For such a meaningful article, the abstract could expanded.  For example, you can add to it defining fathers and recurrence and the typology a nonhomogeneous group of recurrent fathers.

256-257. “Recurrent fathers also appear likely to have, often multiple health vulnerabilities…”,

273-274 “In terms of physical, mental and emotional health across their life course, there was also evidence of cumulative and/or more pronounced levels of problems experienced…”.

It is not quite clear which control group of fathers compared to recurrent fathers. May be “usual” fathers (married, step and so on) have the same problems? What hypotheses did the researchers have?

508-600. In any case, the idea of providing social and psychological assistance to fathers who do not live with their children is very fruitful, both in the interests of the children and the fathers themselves.

632 “In relation to recurrence, we are arguing for the development of services that hold men equally accountable for the safe care of children and avoid positioning women as disproportionately responsible for children’s welfare”. This is a very good idea. Nevertheless, it has repeated many times before in text. Check it, please.

Author Response

We would like to thank Reviewer 2 for their useful comments. We have responded to each specific point below:

For such a meaningful article, the abstract could expanded.  For example, you can add to it defining fathers and recurrence and the typology a nonhomogeneous group of recurrent fathers.

Response: We appreciate the positive comment made about our article. We were not entirely clear of exactly what the reviewer intended in terms of expanding the abstract, but we have revised one sentence (line 11) to give a little more indication of what we are adding to the picture of recurrence, and fathers' experiences of it. We do also make the point about recurrent fathers not being a homogenous group further down, on line 15. We hope this is acceptable.

256-257. “Recurrent fathers also appear likely to have, often multiple health vulnerabilities…”,

273-274 “In terms of physical, mental and emotional health across their life course, there was also evidence of cumulative and/or more pronounced levels of problems experienced…”.

It is not quite clear which control group of fathers compared to recurrent fathers. May be “usual” fathers (married, step and so on) have the same problems? What hypotheses did the researchers have?

Response: In the first example, the sentence the reviewer has identified follows from a statement about how recurrent fathers compare with non-recurrent fathers in the survey sample. For this reason, we feel the sentence is clear/appropriate enough. We hope reviewer 2 might agree.

Response: In the second example, we accept the sentence needs clarification, as we cannot systematically compare our sample of recurrent fathers with either the survey fathers or the wider population of men - although overall our study does show that recurrent fathers are highly marginalised/vulnerable. We have therefore removed the word 'more' from the sentence - line 282.

632 “In relation to recurrence, we are arguing for the development of services that hold men equally accountable for the safe care of children and avoid positioning women as disproportionately responsible for children’s welfare”. This is a very good idea. Nevertheless, it has repeated many times before in text. Check it, please.

Response: We appreciate the need to avoid repetition, but we have re-read the manuscript and could not find examples of where we had previously made this exact point. Across the article, we do point to different contexts where gender difference or gendered responses to fathers and mothers amount to forms of inequality, and we do seek to emphasise that failing to work effectively with fathers amounts to holding mothers overly-responsible. However, we did not feel we had simply repeated the same point multiple times. We would be happy to discuss further or respond to other specific examples of such repetition if need be. Again, we do appreciate the reviewer's general point about weakening a good idea through repetition.